# Walking Stability and Risk of Falls

**DOI:** 10.3390/bioengineering10040471

**Published:** 2023-04-12

**Authors:** Arunee Promsri, Prasit Cholamjiak, Peter Federolf

**Affiliations:** 1Department of Physical Therapy, School of Allied Health Sciences, University of Phayao, Phayao 56000, Thailand; 2Department of Mathematics, School of Sciences, University of Phayao, Phayao 56000, Thailand; 3Department of Sport Science, University of Innsbruck, 6020 Innsbruck, Austria

**Keywords:** gait, neuromuscular control, movement synergy, overground walking, principal component analysis (PCA), largest Lyapunov exponent (LyE)

## Abstract

Walking stability is considered a necessary physical performance for preserving independence and preventing falls. The current study investigated the correlation between walking stability and two clinical markers for falling risk. Principal component analysis (PCA) was applied to extract the three-dimensional (3D) lower-limb kinematic data of 43 healthy older adults (69.8 ± 8.5 years, 36 females) into a set of principal movements (PMs), showing different movement components/synergies working together to accomplish the walking task goal. Then, the largest Lyapunov exponent (LyE) was applied to the first five PMs as a measure of stability, with the interpretation that the higher the LyE, the lower the stability of individual movement components. Next, the fall risk was determined using two functional motor tests—a Short Physical Performance Battery (SPPB) and a Gait Subscale of Performance-Oriented Mobility Assessment (POMA-G)—of which the higher the test score, the better the performance. The main results show that SPPB and POMA-G scores negatively correlate with the LyE seen in specific PMs (*p* ≤ 0.009), indicating that increasing walking instability increases the fall risk. The current findings suggest that inherent walking instability should be considered when assessing and training the lower limbs to reduce the risk of falling.

## 1. Introduction

Falls have been linked to a loss of function and independence in older people, leading to injury-related hospitalizations in the aging population worldwide [1]. They usually occur according to degenerative changes of postural reflex impairment accompanied by the inherent aging process [2]. Approximately one-third of older adults (>65 years) living in the community fall yearly [3], leading to several types of injuries (e.g., pain, soft tissue injuries, fractures, dislocations, and functional impairment [4]) and impacting the quality of life [2]. As previously reported [5,6], several internal risk factors for falling have been reported, e.g., previous history of falls, balance impairment, functional limitations, visual impairment, gait impairment, decreased muscle strength, arthritis, diabetes, pain, using polypharmacy or psychoactive drugs, depression, dizziness, age over 80 years, female sex, and cognitive impairment. Analyzing the main fall risk factors, which is crucial for prevention, has frequently been performed [1,2,4,5].

One of the physical performances necessary for preserving independence and minimizing the risk of falls is the ability to walk successfully and safely on both stable and unstable surfaces [7]. Walking instability has been recognized as one of the leading contributors [5,6] among the several risk factors for falls. Commonly, stability is described as the intrinsic ability of a motor system to retain or recover to its initial condition in the face of internal (e.g., neuromuscular) and external (e.g., environmental) perturbations [8,9]. In this sense, stability measures yield relevant information on the intrinsic noise in motor task performance and directly quantify the performance of dynamic error correction [8,9]. Alternatively, variability measures have also been used to indirectly quantify how stable a person performs locomotion tasks due to inherent noise in the motor tasks or the environment that can bring an individual’s dynamic state closer to their stability limits [8,9,10]. Furthermore, since human movement is believed to result from nonlinear interactions between multiple neuromuscular elements and internal and external factors [11], the largest Lyapunov exponent (LyE), one of the nonlinear methods to assess local dynamic stability, is frequently used to analyze the capacity to manage for small internal or external perturbations in order to maintain functional locomotion (i.e., used to measure walking stability) [9,12,13,14].

In order to complete any given motor activity (e.g., walking), the cooperative contribution of multi-body segments is needed, typically seen as different movement components/synergies forming together to accomplish the task goal [10,15,16]. Principal component analysis (PCA), one of the methods for reducing the number of dimensions, has widely been used on kinematic marker data to extract movement components or synergies, which have been called “principal movements” (PM), from the original, whole postural movements [15,17]. This method helps by minimizing the number of features (i.e., redundancy issues in motor apparatus) needed to finish the given task goal by forming fewer new variables, which still contain the most information regarding how people move or generate motions from the original feature set of postural movements [15,17,18,19]. Moreover, information about the position and acceleration of individual PMs reveals their direct association with system forces and myoelectric activity [20,21], confirming that PCA-based variables have an adequate probability of assessing neuromuscular control of individual movement components/strategies [15,20,21,22]. Regarding local dynamic stability as measured by the LyE, walking stability can be referred to as the neuromuscular system’s ability to manage infinitesimal perturbations during locomotion [9,12,13,14]. Therefore, the LyE applied to individual PM positions can aid in quantifying the stability of individual movement components/strategies that come together to achieve locomotion tasks [10,16,23].

Several functional motor tests have been developed to assess physical performance, since poor physical performance, balance impairment, and gait alterations are among the leading causes of falls in older individuals [24]. When focusing on gait ability, functional motor tests assessing gait ability are commonly used to determine the risk of falling. For example, the Short Physical Performance Battery (SPPB) is a well-established tool for quantifiably assessing the lower extremity physical performance based on three tasks: repeated chair stand, standing balance, and walking speed [25]. Unlike the SPPB, the Gait Subscale of Performance Oriented Mobility Assessment (POMA-G) assesses the quality of walking by considering gait initiation, step length, step height, step symmetry, step continuity, path, trunk movement, and walking stance [26]. The results of these two tests are represented as ordinal scores, ranging from 0 to 12, considered the worst-to-best performance [25,26]. Both tests are reported to accurately discriminate between fallers and non-fallers in a large group of frail older adults [27]. Practically, fall risks are usually predicted using multi-item or functional motor assessment tools [28]. For example, it has been reported that SPPB [29] and POMA-G [30] have the practical ability to predict falls. In this sense, since the ability to maintain stability while walking is critical for avoiding falls, particularly in older adults [31], studying the relationship between falling risk and walking stability by considering movement patterns (i.e., movement strategies) can help to identify individuals who are at higher risk of falling and develop effective interventions to improve walking stability and reduce the falling risk.

In summary, the main purpose of the current study was to determine the correlation between walking stability and the risk of falling. Walking stability was defined in terms of individual PMs’ local dynamic stability (Lyapunov stability), and fall risk was determined by two functional motor tests—SPPB and POMA-G. Since the stability of individual PMs reflects the neuromuscular control of individual movement components or movement synergies [10], it was hypothesized that the correlation between walking stability and the risk of falling would appear in the specific relevant PMs to the gait cycle.

## 2. Materials and Methods

### 2.1. Secondary Data Analysis

The lower-limb kinematic marker data of 43 healthy older adults (36 females and 7 males) used in the current study was derived from a peer-reviewed open-access dataset [32]. All participants had no neurological or musculoskeletal problems concerned with the risk of falling or affecting walking ability. The Mini-Mental State Examination (MMSE) was utilized to assess the mental status (i.e., mental health) to confirm that all participants could understand the experiment protocol and complete the tasks. In addition, two functional motor tests—SPPB and POMA-G—were performed on each participant by an experienced physiotherapist. The Ethics Committee of the Escuela Colombiana de Ingeniería and Clínica Universidad de la Sabana, Colombia, approved the study protocol in accordance with the ethical principles of the Helsinki Declaration, and all participants provided written informed consent before participation, as reported in Caicedo et al. [32]. The participant characteristics are represented in Table 1.

Experimental measurement procedures were detailed and explained in Caicedo et al. [32]. In brief, each participant was equipped with 24 reflective markers, ten at each leg and four around the hip, as shown in Caicedo et al. [32]. The optical motion capture system comprised seven cameras (Vantage V5, Vicon Motion Systems, Ltd., Oxford, UK), with the sample rate set at 100 Hz. Each camera was mounted on a tripod at 1.90 m above the floor. For each walking trial, a C3D file is generated by Nexus movement analysis software, version 2.9.3 (Vicon Motion Systems, Ltd., Oxford, UK), with an accuracy better than 0.3 mm. Each participant was instructed to walk ten times at a self-preferred speed between two points six meters apart, while one researcher walked beside them to ensure their safety during walking. However, the data of the best five walking trials of each participant were provided in the original data article. The current study selected only three walking trials in which all participants walked in the same direction (e.g., walking from point A to point B but not from point B to point A), as checked by running the C3D files for further analysis.

### 2.2. Movement Synergy Extraction

All data processing for the current study was conducted in MATLAB version 2022a (MathWorks Inc., Natick, MA, USA). For each dataset, 16 markers were placed on the main anatomical landmarks (ASIS, PSIS, thigh, knee, tibia, lateral malleolus, heel, and toe) of each leg. These markers gave 48 spatial coordinates (x, y, z), which were interpreted as 48-dimensional posture vectors [15]. Each participant’s kinematic dataset of three walking trials was pre-processed, centered by subtracting the mean posture vector [15], and normalized to the mean Euclidean distance [15] before they were concatenated to form one input matrix (3 trials × 43 participants) for further PCA. Appendix A, an animated stick figure video, shows an example of the original overground walking movement obtained from one female participant.

PCA was carried out with a singular-value decomposition of the covariance matrix through the PManalyzer software [15] to extract all lower-limb kinematic data into a set of orthogonal eigenvectors, which has been called “principal components” (PC*_k_*; *k* indicates the order of movement components). For each orthogonal eigenvector, an animated stick figure called “principal movement” (PM*_k_*), can be created to characterize its movement pattern [15]. The use of the term ”principal” in the variable names denotes that those variables were derived from PCA, of which (*t*) indicates that these variables are functions of time *t* [15]. Furthermore, the actual time evolution (i.e., time series) of each PM is quantified by the PC scores (i.e., principal positions; PP*_k_*(*t*)), which represent the positions in posture space or the vector space spanned by the PC-eigenvectors [15]. In analogy to Newton’s mechanics, PM*_k_*-accelerations (i.e., principal accelerations; PA*_k_*(*t*)), a second-time derivative, can be computed from the PP*_k_*(*t*) based on the conventional differentiation rules [15]. As previously reported in a postural control study [20], PA*_k_*(*t*) have associations with leg myoelectric activity, supporting the idea that PA-based variables could be used to determine the neuromuscular control of individual PM*_k_* [21,33,34,35]. A Fourier analysis was performed on the raw PP*_k_*(t) [35] to detect noise amplification that occurred in the differentiation processes, showing that the highest power resided in a range of frequencies between 2 and 5 Hz, but that the visible power was still seen in the frequency range between 5 and 10 Hz. Hence, the PCA-based time series were filtered with a 3rd-order zero-phase 10-Hz low-pass Butterworth filter before performing the differentiation step. In addition, based on a previous study [15], leave-one-out cross-validation was performed to assess the vulnerability of individual PM*_k_* and the PCA-based dependent variables that change the input data matrix to address validity considerations. In this regard, the current study selected the first five PCs that proved robust to test the hypotheses.

In order to describe the coordinative structure of PM_1–5_, the compositions of overground walking movements were assessed based on their principal position (PP*_k_*(*t*)) and acceleration (PA*_k_*(*t*)) [35]. First, the participant-specific *relative explained variance* of PP*_k_*(*t*) (PP*_k_*_rVAR) was computed to investigate the percentage of the contribution of each PM to the total variance in postural positions, quantifying how important each PM*_k_* is for the overall coordinative movement structures of the overground walking movements [17,33]. Second, the *relative explained variance* of the PA*_k_*(*t*) (PA*_k_*_rVAR) was computed, which quantifies the percentage of the contribution of each PM to the total variance in postural accelerations [20,22,36]. A greater PA*_k_*_rVAR value reflects that a given movement component is performed fast enough to impact accelerations and forces acting in the system [36].

### 2.3. Investigating Walking Stability

Each PP*_k_*(t) was normalized to an individual’s walking speed [23,37]. Then, the participant-specific *largest Lyapunov exponent* (LyE) of PP_1–5_(*t*) or PP*_k_*_LyE was used to investigate walking stability by computing the rate of divergence of close trajectories in state space (i.e., the ability of the motor system to attenuate small perturbations revealed by the divergence of the trajectories in state space) [10,16,23,38].

PP*_k_*_LyE was computed by applying Wolf’s algorithm [39], with the time delay (*τ* = 10) and embedding dimension (m = 4) determined using the average mutual information (AMI) [10,38] and the false nearest neighbor algorithms [40], respectively. A greater PP*_k_*_LyE value indicates the inability of the motor system to reduce infinitesimal perturbations [13], resulting in a greater divergence of state space trajectories. In other words, a higher PP*_k_*_LyE value reflects a lower individual’s walking stability [16,23]. For statistical analysis, the current study used the average of individual PP*_k_*_LyE values calculated from three walking trials.

### 2.4. Statistical Analysis

All statistical analyses were performed using the IBM SPSS Statistics software, version 26.0 (SPSS Inc., Chicago, IL, USA), with the alpha level set at a = 0.05. A Shapiro–Wilk test was used to determine the data’s normality, suggesting using a Spearman’s rho test to determine the correlation between participants’ demographic data (age, BMI, MMSE, walking speed (WS), SPPB, and POMA-G) and individual PP_1–5__LyE. Pearson correlation was used to examine the relationship between individual PP_1–5__LyE. The correlation coefficient (*r*), which varies between −1 and +1, represents the strength of the relationship between the two variables in positive or negative directions, respectively. The absolute correlation (ǀ*r*ǀ) in the range of 0 to 0.4 is interpreted as a weak correlation, 0.4 to 0.8 as a moderate correlation, and 0.8 to 1 as a strong correlation [41].

## 3. Results

### 3.1. Movement Synergies

Table 2 shows the descriptive characteristics of the first five principal movements (PM_1–5_), which together explained 99.9% of the total position variance (PP*_k_*_rVAR) and 70.9% of the acceleration variance (PA*_k_*_rVAR). In addition, the example visualizations of PM_2–5_ are shown in Figure 1.

As shown in Table 2, the highest value of PA*_k_*_rVAR is observed for PM_2_, resembling the swing phase movement, followed by PM_4_, representing ankle and knee flexion and extension movements in the vertical direction; and PM_5_, resembling the mid-stance phase movement, respectively.

### 3.2. Relationship between Walking Stability and Risk of Falls

As shown in Table 3, the main results show that correlations appear in specific pairs of two variables. Regarding the demographic data, the age of participants is negatively correlated with MMSE (*r* = −0.449 (moderate correlation), *p* = 0.003), POMA-G (*r* = −0.450 (moderate correlation), *p* = 0.002), and PP_3__LyE (*r* = −0.306 (weak correlation), *p* = 0.046). The BMI of participants is negatively correlated with SPPB (*r* = −0.355 (weak correlation), *p* = 0.020), but positively correlated with two walking stability variables: PP_2__LyE (*r* = 0.343 (weak correlation), *p* = 0.024) and PP_4__LyE (*r* = 0.506 (moderate correlation), *p* = 0.001). The MMSE value is positively correlated with POMA-G (*r* = 0.379 (weak correlation), *p* = 0.012).

In addition, walking speed is negatively correlated with both two functional motor tests: SPPB (*r* = −0.556 (moderate correlation), *p* < 0.001) and POMA-G (*r* = −0.356 (weak correlation), *p* = 0.019), but is positively correlated with specific walking stability variables: PP_2__LyE (*r* = 0.516 (moderate correlation), *p* < 0.001), PP_4__LyE (*r* = 0.635 (moderate correlation), *p* < 0.001), and PP_5__LyE (*r* = 0.428 (moderate correlation), *p* = 0.004).

Regarding the functional motor tests, SPPB negatively correlates with the specific walking stability variable, PP_4__LyE (*r* = −0.402 (moderate correlation), *p* = 0.008). In addition, POMA-G negatively correlates with the specific walking stability variables: PP_4__LyE (*r* = −0.417 (moderate correlation), *p* = 0.005) and PP_5__LyE (*r* = −0.396 (weak correlation), *p* = 0.009).

Moreover, correlations within the individual PP*_k_*_LyE are observed in the specific pairs of PP*_k_*_LyE. Specifically, PP_2__LyE is positively correlated with PP_4__LyE (*r* = 0.718 (moderate correlation), *p* < 0.001), and PP_5__LyE (*r* = 0.443 (moderate correlation), *p* = 0.003). PP_4__LyE is positively correlated with PP_5__LyE (*r* = 0.386 (weak correlation), *p* = 0.011).

## 4. Discussion

The current study determined the correlation between walking stability and fall risk in healthy older adults. Walking stability defined in terms of local dynamic stability was assessed through the largest Lyapunov exponent (LyE) of individual movement components or movement synergies (i.e., called “principal movements,” PMs) extracted by applying principal component analysis (PCA) to overground walking movements. The fall risk was determined by two functional motor tests—the Short Physical Performance Battery (SPPB) and the Gait Subscale of Performance-Oriented Mobility Assessment (POMA-G). The main results show that negative, small-to-moderate correlations between PPk_LyE and two functional motor tests (SPPB and POMA-G) appear in the specific PMs, suggesting that the lower the PP_4_-Lyapunov stability, the greater the risk of falling. Based on the empirical findings, two main points can be discussed.

First, the lower performance of the lower extremities possibly influences walking instability, especially in movement components resembling the ground contact phases of the gait cycle (PM_4–5_). Walking instability can be caused by a degenerative change in the lower-limb muscle–tendon neuromechanics (e.g., a decline in muscle strength [42] and a degenerative muscle [43] and tendon [44] property), which usually happens as a normal part of the inherent aging process [45]. This degenerative physical decline could make it harder to control body weight while walking [46]. For example, in the PM_4_, which represents the ankle and knee flexion and extension movements, the declining calf muscle strength (e.g., the gastrocnemius, as the two joint muscles associated with both ankle and knee movements) may be involved in the instability of this movement component. A previous review article reported age-related declines in the contribution of the Achilles tendon in recoiling to ankle power output during walking, leading to an increase in the metabolic cost of walking because of less economical calf muscle contractions and increased work of the proximal joint (e.g., the hip joints) [44]. This point is of interest and may need further analysis. In addition, in the PM_5_, which resembles the mid-stance phase, the hamstring muscles are an essential group of muscles that play the main role in the weight-bearing and takeoff phases of the gait cycle for three functions [46]: (I) decelerating the knee extension through an eccentric contraction at the end of the swing phase to stabilize the weight-bearing knee dynamically; (II) facilitating the hip extension through an eccentric contraction at foot strike to stabilize the weight-bearing leg; and (III) supporting the gastrocnemius muscles through an eccentric contraction in extending the knee during the takeoff phase.

Second, since SPPB [29] and POMA-G [30] have the potential to predict the risk of falls in terms of measuring lower-limb physical performance, walking instability should be considered a potential fall risk. Although SPPB and POMA-G assess lower limb performance, they focus on different aspects. For example, the SPPS measures lower-limb performance in terms of time spent performing standing balance, walking speed, and chair stand tests [47]. Unlike the SPPB, the POMA-G focuses on the quality of walking, e.g., the ability of gait initiation, step length, step height, step symmetry, step continuity, walking path, trunk movement, and walking stance [26]. In this sense, the SPPB is one of the functional tests practically used to assess lower extremity strength [29] and used as a predictor of mortality in older adults by all causes [47].

Regarding the characteristics of participants, age has negatively correlated with the MMSE and POMA-G, indicating possible cognitive [48] and gait [26] impairments that may occur with advancing age. The BMI negatively correlates with SPPB, indicating that individuals with increasing body mass relative to height may have lower limb muscle strength [29] and physical performance [25]. The MMSE positively correlates with the POMA-G, indicating that individuals with possible cognitive impairment [48] may have been associated with gait impairments [26]. In addition, walking speed negatively correlates with both SPPB and POMA-G, indicating that reduced walking speed reflects decreased physical performance in individuals. Moreover, walking speed has a positive correlation with walking instability, reflecting that increased walking speed increases walking instability. Based on these findings, it is suggested that individuals with an advancing age, an increasing BMI, a decreasing MMSE, and a reduced walking speed are associated with lower physical performance, possibly leading to an increased risk of falling.

When considering the correlation among the PP*_k_*_LyE, a positive interrelationship between the walking stability variables is observed between PM_2_ (PP_2__LyE) and PM_4–5_ (PP_4–5__LyE), indicating that the higher the instability of the swing phase, the greater the instability of the contact ground movements of the two legs. In addition, a positive interrelationship between PM_4_ (PP_4__LyE) and PM_5_ (PP_5__LyE) indicates that the higher the instability of the swing phase, the greater the instability of the mid-stance phase. Although these three movement components (PM_2,4–5_) are movement components that are small in positional amplitude (PP_2,4–5__rVAR), they are performed fast enough (PA_2,4–5__rVAR) to influence accelerations considerably, and thus forces acting in the system [36]. In this sense, fall prevention programs should take into account how unstable a person is during both the swing and stance phases of a gait cycle.

In terms of practical application, the current study suggests that reducing walking Lyapunov stability, specifically in the ground contact movement components (PP_4–5_), should be considered for fall prevention and rehabilitation, for which task-specific gait training to improve neuromuscular control of the lower extremities is recommended. For instance, the three subtasks of the SPPB—chair stand, standing balance, and walking speed [25]—can be applied as an exercise or training for fall prevention. Furthermore, exercising or training to improve walking quality by considering the POMA-G components—gait initiation, step length, step height, step symmetry, step continuity, walking path, trunk movement, and walking stance [26]—is of interest and can be practical in clinical settings.

### Limitations and Future Study

One limitation of the current study was that only the lower limb movements provided by an open-access dataset were analyzed. Therefore, for future research, whole-body movement analysis is suggested since the effective contribution of all the body segments is required for achieving the given task goal [21], representing that the neuromuscular system controls posture and movement through multiple muscles that produce relative movements between multiple body segments [20]. Another limitation was that the characteristics of participants enrolled in the current study were not generalized, but mostly female. In this regard, considering the impact of the sexes [5,6] or investigating the age-related differences in walking stability is suggested for future research.

Since, in the current study, only the correlation test was performed to study the relationship between walking stability and the risk of falling, applying the regression analysis focused on modeling the relationship may be of interest. Moreover, the risk of falls is considered highly correlated to lower extremity muscle strength and joint moments [49,50], usually observed in frail, older adults [51] or individuals with neurological or musculoskeletal impairments [52]. Therefore, encouraging the collection of kinematics combined with kinetic or electromyographic (EMG) data is suggested [20], since it is highly informative and may offer insights into net muscle forces acting at the joints, especially during periods of the single support phase of the gait cycle.

## 5. Conclusions

In healthy older adults, the negative small-to-moderate correlations are observed between the Lyapunov instability of specific movement components (i.e., principal movements, PMs) extracted from the lower limb movements during overground walking with self-selected speed and the potential risk of falls assessed by two functional motor tests—the Short Physical Performance Battery (SPPB) and the Gait Subscale of Performance-Oriented Mobility Assessment (POMA-G), indicating the higher the LyE, the lower the physical performance with possibly increased risk of falling. Based on the current findings, it is, therefore, suggested that the inherent impacts of walking (Lyapunov) stability should be considered for fall investigation, prevention, and rehabilitation, not particularly in healthy older adults but also in frail, older adults and individuals with neurological or musculoskeletal impairments, possibly increasing the risk of falls.

## Figures and Tables

**Figure 1 bioengineering-10-00471-f001:**
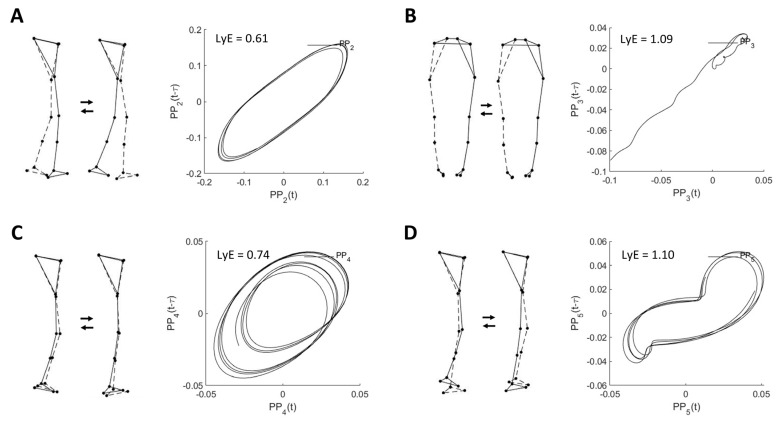
Example visualizations of (**A**) PM_2_, (**B**) PM_3_, (**C**) PM_4_, and (**D**) PM_5_ extracted from the overground walking movement and their corresponding space-time representation for computed largest Lyapunov exponent (LyE) of individual PP*_k_*. Note: LyE data are derived from the first trial of one female participant. The dashed line indicates the left limb. Only PM_3_ is shown in the back view.

**Table 1 bioengineering-10-00471-t001:** Descriptive characteristics of participants (*n* = 43).

	Min	Max	Mean	SD
Age (years)	54.0	87.0	69.8	8.5
Mass (kg)	41.8	104.4	67.6	11.2
Height (m)	1.4	1.7	1.6	0.1
Body Mass Index (kg/m^2^)	17.4	40.3	27.8	4.5
MMSE	22.0	30.0	26.6	2.5
SPPB	5.0	12.0	9.8	1.7
POMA-G	8.0	12.0	10.2	0.8
Walking speed (m/s)	0.6	1.2	0.8	0.2
Number of falls in the last month (time)	0	1	0.1	0.3

**Table 2 bioengineering-10-00471-t002:** The relative explained variances (mean ± SD) of the principal positions (PP*_k_*_rVAR) and the principal accelerations (PA*_k_*_rVAR) of the first five principal movements (PM_1–5_), amended with a qualitative description of the main features of each movement component. Note: *k* indicates the order of principal movements, and animated stick figures of PM_2−5_ are represented in Appendix A.

PM*_k_*	Descriptive Characteristics	PP*_k_*_rVAR	PA*_k_*_rVAR
1	Movements of the lower extremities in the direction of walking	98.91 ± 0.33	4.90 ± 1.12
2	Resemble swing phase movement of the gait cycle: the anti-phase lower-limb movements in the anteroposterior direction	0.90 ± 0.25	31.67 ± 2.94
3	Movements of the lower extremities in the mediolateral direction (i.e., mediolateral sway) combined with anti-phase knee flexion and extension movements in the vertical direction	0.07 ± 0.12	0.43 ± 0.17
4	Both ankle and knee flexion and extension movements in the vertical direction	0.05 ± 0.01	24.65 ± 1.95
5	Resemble the mid-stance phase movement of the gait cycle: the anti-phase lower-limb movements in the vertical direction	0.04 ± 0.01	9.22 ± 1.95

**Table 3 bioengineering-10-00471-t003:** Correlation coefficients (*r*) between participants’ demographic data (age, BMI, MMSE, SPPB, and POMA–G) and individual PP_1–5__LyE. Note: *p*-values smaller than 0.05 are printed in bold (*n* = 43; * *p* < 0.050; ** *p* < 0.01; and *** *p* ≤ 0.001 (two-tailed)).

Variable	1	2	3	4	5	6	7	8	9	10	11
1. Age	1										
2. BMI	−0.063	1									
3. MMSE	**−0.449 ****	−0.290	1								
4. WS	0.242	0.206	−0.122	1							
5. SPPB	−0.205	**−0.355 ***	0.142	**−0.556 *****	1						
6. POMA-G	**−0.450 ****	−0.051	**0.379 ***	**−0.356 ***	0.146	1					
7. PP_1__LyE	0.178	0.173	−0.086	−0.001	−0.100	0.043	1				
8. PP_2__LyE	0.102	**0.343 ***	−0.145	**0.516 *****	−0.164	−0.249	0.032	1			
9. PP_3__LyE	**−0.306 ***	0.145	0.030	0.099	−0.097	0.003	0.066	0.075	1		
10. PP_4__LyE	0.145	**0.506 *****	−0.186	**0.635 *****	**−0.402 ****	**−0.417 ****	0.160	**0.718 *****	0.050	1	
11. PP_5__LyE	0.266	0.091	−0.097	**0.428 ****	−0.046	**−0.396 ****	0.021	**0.443 ****	−0.056	**0.386 ***	1

## Data Availability

Not applicable.

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
