# Peer review of "Walking Stability and Risk of Falls"

_bioengineering, 2023, doi:10.3390/bioengineering10040471_

Round 1

Reviewer 1 Report

The purpose of this study was to determine correlations between aspects of stability while walking with falling risk.  The former were derived by means of principal component analysis while the latter came from triangulating two functional motor tests.  Participants also completed a cognitive test.

The manuscript is well-written and the research design is simple and straightforward.  Descriptions of how the walking kinematic data were manipulated were challenging to weed through, but this was necessary to demonstrate competence in the technique.

While I think the study was worthwhile and will be of interest to those studying balance and stability along with falls prevention, in my opinion, this technique would likely generate even more relevant information had the authors also included walking kinetic data, i.e., joint moments of force and powers, in the analysis.  The limitation of kinematics, which are relatively easy to obtain, is that they are the effects of the kinetics, i.e., the causes of the motion being observed.  The risk of falls is highly correlated to lower extremity muscle strength, and I believe including joint moments, which offer insights into net muscle forces acting at the joints, especially during periods of single support, would be highly informative.  I strongly encourage the authors to collect both kinematic and kinetic data in their next study (and, perhaps, EMG in another subsequent study).

The formatting of Table 2 needs to be fixed so that each PMk number lines up with its respective numerical data and the first line of the descriptive characteristics. Hence, it is clear each characteristic lines up with its PMk number. 

Author Response

Response to Reviewer 1 Comments

Point 1: The purpose of this study was to determine correlations between aspects of stability while walking with falling risk.  The former were derived by means of principal component analysis while the latter came from triangulating two functional motor tests.  Participants also completed a cognitive test.

The manuscript is well-written and the research design is simple and straightforward.  Descriptions of how the walking kinematic data were manipulated were challenging to weed through, but this was necessary to demonstrate competence in the technique.

Response 1: thank you very much for your time and effort in reviewing our manuscript. We appreciate all the valuable, constructive comments and suggestions. We hope that we correctly understand all the suggestions and improve the revised manuscript in the correct way. Please see our point-by-point responses below and the changes in the revised manuscript:

Point 2: While I think the study was worthwhile and will be of interest to those studying balance and stability along with falls prevention, in my opinion, this technique would likely generate even more relevant information had the authors also included walking kinetic data, i.e., joint moments of force and powers, in the analysis.  The limitation of kinematics, which are relatively easy to obtain, is that they are the effects of the kinetics, i.e., the causes of the motion being observed.  The risk of falls is highly correlated to lower extremity muscle strength, and I believe including joint moments, which offer insights into net muscle forces acting at the joints, especially during periods of single support, would be highly informative.  I strongly encourage the authors to collect both kinematic and kinetic data in their next study (and, perhaps, EMG in another subsequent study).

Response 2: thank you very much for your interesting, constructive suggestions. We agree with your opinion. In the current study, we analyzed the public datasets that did not provide kinetic data. However, your suggestion is beneficial for future research, so we added this suggestion to the discussion.

Point 3: The formatting of Table 2 needs to be fixed so that each PMk number lines up with its respective numerical data and the first line of the descriptive characteristics. Hence, it is clear each characteristic lines up with its PMk number. 

Response 3: thank you very much for the suggestion. We improved the table heading and rearrange the table column accordingly.

Reviewer 2 Report

1.The abstract is too long and needs to further refinement.

2. The motivation for the research on Walking Stability and Risk of Falls in this article is insufficient and needs to be hightlighted.

3. The figures  are not clear and requires a higher resolution image.

4. It is necessary to further explain the experimental population and increase the test results for different age groups.

5. The format of reference is not uniform.

Author Response

Response to Reviewer 2 Comments

Point 1: 1.The abstract is too long and needs to further refinement.

Response 1: thank you very much for the comment. In the revised manuscript, we managed the Abstract section so that the words in this part are 199 words as in a range of the number of words recommended by the journal.

Point 2: 2. The motivation for the research on Walking Stability and Risk of Falls in this article is insufficient and needs to be hightlighted.

Response 2: thank you very much for the comment. We tried our best to highlight the motivation for the current research in the Introduction section of the revised manuscript.

Point 3: 3. The figures are not clear and requires a higher resolution image.

Response 3: thank you very much for the comment. We checked and tried to increase the resolution of the figure (768 dpi). I am unsure whether the low resolution of the figure occurred due to the PDF printing process. However, with the revised submission, we attached the figure file accordingly.

Point 4: 4. It is necessary to further explain the experimental population and increase the test results for different age groups.

Response 4: thank you very much for the comments. The current study used the online public dataset to answer our research questions about whether walking stability correlates with the fall risk estimated by two functional motor tasks. So, we did not test the effects of age, but we appreciate your suggestions and may apply them to our future research.

Point 5: 5. The format of reference is not uniform.

Response 5: thank you very much for the comment. In the current study, we used Mendeley and selected the style “Bioengineering” provided by the software to operate the references. However, I did update all the references and hope they are in a suitable format.

Reviewer 3 Report

Abstract - concise, accurate, pertinent.

Introduction - states the problem, provides comprehensive background and rationale/need for study, concludes with explicit purpose statement, including hypotheses

Materials and Methods

line 103 - suggest  "...older, healthy adults..."

Table 1 - should be: Weight (N), Weight (lb) or Mass (kg).

line 126 - Please describe the criteria for selecting the 3 trials.

Results

text and Table 3 - The largest r value = 0.718, which gives an r^2 ~ 0.50. Is this case only 50% of the variance can be explained. Therefore, although you have many statistically significant r values, their r^2 values are quite low. Consider this in your discussion. Frankly, I believe that using r^2 is more valuable because it forces one to be more conservative in making conclusions/implications.

Discussion

line 262 - Consider also the influence of degenerative joint disease on muscle, eg, Yasuda et al. Arthroplasty (2022) 4:23 https://doi.org/10.1186/s42836-022-00126-7. So, there is a relationship between joint disease and muscle-tendon degeneration, and the cause is a "chicken or the egg" dilemma.

limitations - You do not have a control group, ie, younger age. For example, the local stability measures may occur at different PMs.

Conclusions - Consider telling the reader again what the most important relationships are instead of claiming that "the current study highlights…"

References - strong

Author Response

Response to Reviewer 3 Comments

Point 1: Abstract - concise, accurate, pertinent.

Introduction - states the problem, provides comprehensive background and rationale/need for study, concludes with explicit purpose statement, including hypotheses

Response 1: thank you very much for your time and effort in reviewing our manuscript. We appreciate all the valuable, constructive comments and suggestions. We hope that we correctly understand all the suggestions and improve the revised manuscript in the correct way. Please see our point-by-point responses below and the changes in the revised manuscript:

Point 2: Materials and Methods

line 103 - suggest  "...older, healthy adults..."

Table 1 - should be: Weight (N), Weight (lb) or Mass (kg).

line 126 - Please describe the criteria for selecting the 3 trials.

Response 2: thank you very much for the suggestions.

  • We added the suggested phrase in the section “Participants” accordingly.
  • In Table 1, we replaced the word “Weight (kg)” with the word “Mass (kg)”.
  • In line 126, we rewrote the sentence explaining the criteria for selecting only three trials that all participants walked in the same direction (e.g., walking from point A to B but not from point B to A). For example, most participants walked from A to B and then from B to A. Hence, only three walking trials were selected, since the original data article provided five walking trials. If we pooled the data from different walking starting directions, they affected the results, which is incorrect.

Point 3: Results

text and Table 3 - The largest r value = 0.718, which gives an r^2 ~ 0.50. Is this case only 50% of the variance can be explained. Therefore, although you have many statistically significant r values, their r^2 values are quite low. Consider this in your discussion. Frankly, I believe that using r^2 is more valuable because it forces one to be more conservative in making conclusions/implications.

Response 3: thank you very much for pointing out an important issue. In the revised manuscript, we added the strength of correlation value in the results and in the discussion sections and tried our best to rewrite the discussion part to make it clearer. To be honest, while analyzing the data, we forgot the regression analysis and focused only on the correlation test. Thank you again for pointing out this analysis, however, we would like to present the results of the correlation test. The regression analysis is of interest to our future research.

Point 4: Discussion

line 262 - Consider also the influence of degenerative joint disease on muscle, eg, Yasuda et al. Arthroplasty (2022) 4:23 https://doi.org/10.1186/s42836-022-00126-7. So, there is a relationship between joint disease and muscle-tendon degeneration, and the cause is a "chicken or the egg" dilemma.

Response 4: thank you very much for your suggestion. It is an interesting point that we added to the revised manuscript.

Point 5: limitations - You do not have a control group, ie, younger age. For example, the local stability measures may occur at different PMs.

Response 5: thank you very much for the comment. We do not have a control group. In this sense, we add the suggestion for future research, e.g., studying the age effects, according to the point you raised. It is an interesting idea.

Point 6: Conclusions - Consider telling the reader again what the most important relationships are instead of claiming that "the current study highlights…"

Response 6: thank you very much for your comment. We rewrote the conclusion section to present the main findings accordingly.

Point 7: References - strong

Response 7: thank you very much.

Round 2

Reviewer 3 Report

Thank you for adequately addressing my questions.